# Amyloid Fibrils of Stefin B Show Anisotropic Properties

**DOI:** 10.3390/ijms24043737

**Published:** 2023-02-13

**Authors:** Matjaž Žganec, Ajda Taler Verčič, Igor Muševič, Miha Škarabot, Eva Žerovnik

**Affiliations:** 1Department of Biochemistry and Molecular and Structural Biology, Jožef Stefan Institute, Jamova 39, 1000 Ljubljana, Slovenia; 2Faculty of Mathematics and Physics, University of Ljubljana, Jadranska 19, 1000 Ljubljana, Slovenia; 3Jožef Stefan International Postgraduate School, Jamova 39, 1000 Ljubljana, Slovenia; 4Institute of Biochemistry and Molecular Genetics, Faculty of Medicine, University of Ljubljana, Vrazov trg 2, 1000 Ljubljana, Slovenia; 5Department of Condensed Matter Physics, Jožef Stefan Institute, Jamova 39, 1000 Ljubljana, Slovenia

**Keywords:** nano-sized object, bundle of amyloid fibrils, birefringence, intrinsic fluorescence, label-free imaging

## Abstract

Human stefin B, a member of the cystatin family of cysteine protease inhibitors, tends to form amyloid fibrils under relatively mild conditions, which is why it is used as a model protein to study amyloid fibrillation. Here, we show for the first time that bundles of amyloid fibrils, i.e., helically twisted ribbons, formed by human stefin B exhibit birefringence. This physical property is commonly observed in amyloid fibrils when stained with Congo red. However, we show that the fibrils arrange in regular anisotropic arrays and no staining is required. They share this property with anisotropic protein crystals, structured protein arrays such as tubulin and myosin, and other anisotropic elongated materials, such as textile fibres and liquid crystals. In certain macroscopic arrangements of amyloid fibrils, not only birefringence is observed, but also enhanced emission of intrinsic fluorescence, implying a possibility to detect amyloid fibrils with no labels by using optical microscopy. In our case, no enhancement of intrinsic tyrosine fluorescence was observed at 303 nm; instead, an additional fluorescence emission peak appeared at 425 to 430 nm. We believe that both phenomena, birefringence and fluorescence emission in the deep blue, should be further explored with this and other amyloidogenic proteins. This may allow the development of label-free detection methods for amyloid fibrils of different origins.

## 1. Introduction

The amyloid state is an alternative, usually disease-related conformation of proteins, formed by a cross-β structure, as opposed to the native and unfolded state. The presence of cross-β-sheet structures, in which the β-strands are perpendicular to the fibril axis and the β-sheet is parallel to the fibril axis, confirms amyloid-like aggregates. The X-ray diffraction pattern shows two characteristic reflections at 4.7 Å and 10 Å, corresponding to the distance between the β-strands in the β-sheet and the distance between the β-sheets, respectively [1]. Their structure allows the binding of specific dyes, such as Congo red and thioflavin T (ThT). In addition, transmission electron microscopy (TFE) shows that amyloid fibrils usually consist of two to four filaments that are helically twisted or laterally connected [1,2,3]. Regular arrangements of amyloid fibrils in helically twisted ribbons and elongated bundles also occur in other proteins, such as the smooth muscle protein titin [4]. A recent review paper summarizes most of the general properties of amyloid fibrils, including the tendency to form massive, lattice-like bundles [5]. Several forces that control the interaction between fibrils are discussed [5].

The structure of amyloid fibrils is determined by hydrophobic [6] and aromatic–aromatic ring interactions [7], but also by electrostatic forces [8]. Driven by surface forces amyloid fibrils form quaternary structures of different morphology [5]. The forces that make fibrils stick together to form ribbons are surface adhesion (Van der Waals), while electrostatic forces also play an important role, as the regular repetition of charges along amyloid fibrils keeps them apart due to electrostatic repulsion [8]. Regardless, in amyloid fibrils a regular network of hydrogen bonds between β-strands forms an extended β-sheet structure.

It was predicted that the rings of the side chains of the aromatic amino acids in the amyloid structure are subject to stacking interactions [9], and a later analysis of the crystal structures in Protein Data Bankindeed confirmed the importance of edge-to-edge attractive forces in addition to parallel stacking of the aromatic rings [10]. However, the role of specific aromatic amino acid side chains in amyloid formation may vary [11] and the importance of ring-to-ring stacking interactions has not always been confirmed [12]. Speculation remains that the regular repetition of aromatic ring–ring electron couplings may lead to entanglement-like phenomena in supramolecular amyloid assemblies, as in other structured protein assemblies, e.g., tubulin and myosin cytoskeleton [13].

Many biophysical properties of amyloid fibrils are in common with peptide nanotubes [13] and some with liquid crystals [14]. It has been shown that amyloid fibrils can form ordered liquid crystalline structures that are birefringent and, therefore, can be observed by polarized light microscopy. Unlabeled protein fibrils exhibit anisotropic properties, such as a nematic liquid crystalline phase, which has been demonstrated for hen lysozyme [14]. Another optical technique that can be used to visualize structural protein arrays is second harmonic generation microscopy, which is based on a nonlinear optical effect and has been used to visualize cell and tissue structures. In addition, highly ordered and birefringent arrays, such as myosin and tubulin, have been shown to produce large second harmonic generation signals without the need for exogenous labels [15]. This form of microscopy has several advantages, such as lower photobleaching and less phototoxicity [16]. Such biological materials are promising candidates for nonlinear optical nanomaterials that can be integrated into nanophotonic devices [17].

Another special optical property is photoluminescence, in which blue peaks of light are generated in both nanotubes [18,19] and amyloid fibrils [20]. The blue photoluminescence peak is thought to arise from a network of hydrogen bonds, connecting the β-strands in the β-sheet structure of the amyloid-like nanofibrils [18]. It is speculated that the hydrogen bonds cause low-energy electron transitions [20,21], emitting blue visible fluorescence.

Another feature of peptide nanotubes [13], which may also be present in amyloid fibrils, is enhanced fluorescence emission. It has been observed that amyloid-forming proteins that possess many aromatic amino acid residues exhibit enhanced fluorescence emission when excited at 280 nm. This was demonstrated for insulin and β-lactoglobulin, which exhibited enhanced intrinsic fluorescence with emission spectra that were dependent on the excitation wavelength. In addition, a new deep blue autofluorescence (dbAF) peak was observed at approximately 435 nm for β-lactoglobulin in its amyloid state [22].

We have an interesting model system of recombinant proteins, human stefins A and B (of the cystatin family), which serve as model proteins for the study of amyloid fibrils [23,24]. The mechanism of stefin B fibril formation has been studied in detail [25,26,27] and is consistent with that of many other amyloid-forming proteins. Along the sigmoidal progression of the reaction curve, we have observed various morphologies imaged by both atomic force microscopy (AFM) and TEM [25,26,27]. Fibrils can be obtained from the less stable stefin B at either pH 4.8, 25 °C or pH 7.5, 50 °C; the addition of 9–10% (*v*/*v*) three-fluoroethanol (TFE) can accelerate fibrillation to a reasonable time frame of days [25]. In this brief report, we present data on anisotropic properties of amyloid fibril bundles formed by human stefin B that have not been measured previously. We also explore the enhanced dbAF.

## 2. Results

We prepared amyloid fibrils from human stefin B according to two protocols at pH 4.8 and pH 7.5—as described previously [25]. The kinetics of the reaction were followed by ThT fluorescence [25]. The reaction plateau was reached at pH 7.5 after 25 h. Then, the mature fibrils were analyzed by AFM and TEM (the AFM results in Figure 1 are adapted from our previous work [25]). It can be seen that the fibrils obtained at pH 7.5 are larger (both in width and height) than those obtained at pH 4.8. Therefore, only fibrils obtained at pH 7.5 were used for optical microscopy. As can be seen in Figure 2, birefringent, single elongated bundles with a size of about 100 µm × 15 µm are formed. It must be admitted that we did not detect such a massive bundle using either AFM or TEM in our previous work [25]. One reason for this could be the lower protein concentration used previously.

The polarized microscopy image of the sample prepared pH 7.5 with a protein concentration of 1.5 mM clearly shows that the bundle of amyloid fibrils is birefringent as light passes through the bundle between crossed polarizers (Figure 2A). The birefringence is a consequence of the liquid crystalline alignment of the fibrils, with the local alignment of the optical axis in the bundle determined by the preferred direction of the aligned fibrils [14]. By inserting the full-wave retardation plate (red wave plate), the local alignment of the optical axis and the fibrils inside the bundle can be seen (Figure 2B–D). Here, the red color corresponds to the black color on the crossed polarized image, while the yellow and blue colors correspond to the white colour. The yellow color represents the clockwise rotation of the optical axis from the horizontal and the blue color represents the counterclockwise rotation from the horizontal. The bundle appears blue in Figure 2B so that the fibrils are oriented approximately perpendicular to the long axis of the bundle. When the sample is rotated in the opposite direction, the bundle appears yellow as expected (Figure 2C). Finally, we rotate the bundle along the vertical axis and it is clearly observed that the optical axis is not exactly perpendicular to the long axis of the bundle, but is slightly tilted from the horizontal. By rotating the sample, we estimate that the tilt angle is about 15° from horizontal (Figure 2D). Such alignment of optical axes in fibrils suggests that macroscopic bundles consist of helically twisted fibril ribbons. Not all bundles exhibit such a regular structure—the folded bundle in Figure 3, for example, is not perfectly ordered.

Next, fluorescence emission spectra of stefin B at pH 7.5 were recorded in the range from 200 to 900 nm, with excitation wavelengths between 220 and 380 nm in steps of 20 nm (Appendix A). We compared the fluorescence emission spectra of stefin B in its native state (Appendix A) and in the form of amyloid fibrils (Appendix A). Figure 4 shows fluorescence emission spectra excited at one chosen wavelength, either 280 nm or 330 nm. Spectra of the native protein were measured first, then of an intermediate after the process of heating and of amyloid fibrils upon TFE addition (procedure in Methods). When the native protein is excited at 280 nm, a characteristic peak of intrinsic fluorescence is observed at 303 nm, which is expected for tyrosines in natively folded conformation. As can be seen in Figure 4A, heating to 50 °C alone does not change the native emission peak at 303 nm, while the transition to amyloid fibrils shifts the tyrosines to a non-native environment. Indeed, with amyloid fibrils, we observe a shift of the intrinsic tyrosine fluorescence to a broader peak centered at 356 nm, which is characteristic of unfolded proteins. Due to the turbidity, the exact protein concentration in the fibrillar state is difficult to determine, making it impossible to assess whether the intrinsic fluorescence is enhanced due the concentration difference or not.

Upon excitation from 320 nm onwards, we observe an additional peak of dbAF at around 425–430 nm, both for the native protein and for amyloid fibrils (Appendix A), which has been detected in some other amyloid-forming proteins [20,28] as well. The native state of stefin B at pH 7.5 has a certain amount of higher oligomers [25] that likely produce dbAF. In accordance, we have observed different amounts of autofluorescence at 425 nm with different protein batches, likely dependent on the freeze–thaw cycles (not shown). To avoid the influence of the batches, we treated one sample—as described already for excitation at 280 nm, with temperature, after which we added TFE. We recorded fluorescence emission spectra from 345 to 545 nm, by using excitation wavelength of 330 nm. The results in Figure 4B show that the peak at 425 nm increases with heating to 50 °C. When amyloid fibrils appear upon TFE addition, the peak disappears and is broader (Figure 4C). The results (Figure 4B,C) can be explained as if dbAF appearing after heating would be due to still native conformation (which manifests as the 303 nm peakin Figure 4A). As we know that during amyloid fibrils formation (end of the lag phase and the growth phase) higher oligomers form, the fluorescence emission peak at 425 nm might be due to one such transient species. This is in accordance with results of some other authors as will be discussed.

## 3. Discussion

Properties of nematic phases similar to those of liquid crystals have also been observed in other amyloid-forming proteins, e.g., hen egg-white lysozyme [14]. More recently, studies of other amyloid-forming protein models have been reported [29]. Amyloid fibrils can self-assemble into giant bundles with different morphologies. However, intrinsic birefringence as such has not been described for any of the amyloid-forming proteins to the best of our knowledge. Applications for intrinsic birefringence of regular arrangements of amyloid fibrils can be found in label-free detection of biological samples or in the use of such materials in nanotechnology and biomedicine as sensors or scaffolds [30].

Another potentially useful hallmark of label-free amyloid fibrils is the red shift in emission, known as the “red edge excitation shift (REES)”, which is thought to occur in proteins with many aromatic amino acid residues. However, the enhanced dbAF in fibrillar and monomeric proteins unrelated to aromatic amino acids has been observed previously in several amyloid-forming proteins [20,28]. Various suggestions have been made to explain this phenomenon, such as hydrogen bonding in amyloid fibrils or enhanced carbonyl-based fluorescence, which is already present in amino acids and other carbonyl-containing compounds [28].

In the stefin B study, we observed that the dbAF at 425 nm does not depend on monomers, dimers, or tetramers, but rather on the protein batch, i.e., the number of previous freeze–thaw cycles (not shown), suggesting higher oligomers as the entity responsible for this property. Thus, our experiments confirm that the dbAF is attributable to the early stages of stefin B fibrillation (stage of protofibrils), even before they tend to bind ThT, and that it could be an important detection method for initial states [31,32]. If blue fluorescence emission is indeed more pronounced in the prefibrillar state than in mature fibrils or in the native state, this could be a way to detect such, and presumably most, toxic entities without the need for labels in cell preparations or in living cells.

## 4. Methods

### 4.1. Protein Production and Purification

Recombinant human stefin B was produced in *E. coli* and purified as previously described [27].

### 4.2. Amyloid Fibrillation, ThT Fluorescence

Stefin B amyloid fibrils were obtained in two different buffer conditions: either at pH 4.8 (0.1 M sodium acetate, 0.15 M NaCl, 9% (*v*/*v*) TFE, 25 °C, 300 rpm) or at pH 7.5 (0.02 M ammonium acetate, 0.15 M NaCl, 9% (*v/v*) TFE, 50 °C, 300 rpm). Protein concentration was 34 μM. Aliquots were taken at different time points and ThT fluorescence was measured to follow the fibrillation process. At the plateau of the fibrillation reaction, i.e., after 25 h at pH 7.5 or 120 h at pH 4.8, respectively, the sample was prepared for atomic force microscopy—the results were taken from our previous study [25].

For this study, we incubated the 10-fold concentrated protein at 0.3 mM in 0.01 M phosphate buffer, pH 7.5, 0.15 M NaCl, 10% (*v*/*v*) TFE at 50 °C, and 300 rpm. Fibrillation was followed by ThT fluorescence by taking aliquots at different time points (not shown). At the plateau of the reaction, after 24 h, the sample was concentrated to 1.5 mM stock solution with Amicon 10 MWCO. For further analysis, the sample was diluted to various concentrations ranging from 0.3 mM to 1.5 mM. Fibrils were analyzed by polarized optical microscopy.

ThT was dissolved in buffer pH 7.5 (0.025 M phosphate buffer, 0.1 M NaCl) to an absorbance of 0.66 at 416 nm. From the incubation mixture, 50 μL of sample was removed and added to 570 μL of ThT solution. Fluorescence emission spectra excited at 440 nm were recorded from 455 to 600 nm.

### 4.3. Intrinsic Fluorescence

Intrinsic emission spectra (excited at 280 nm) were recorded using a Perkin–Elmer luminescence spectrometer, model LS 50 B. Both native stefin B in solution and the sample containing stefin B amyloid fibrils were prepared at a concentration of 17 μM in a 0.5 cm quartz cuvette. It should be noted that the concentration of fibrils is less reliable than that of soluble protein. Fluorescence emission spectra were measured from 250 to 50 nm, excited from 220 to 280 nm, in steps of 20 nm. We also measured fluorescence emission spectra from 350 to 550 nm, excited from 300 to 380 nm, in steps of 20 nm.

Additional measurements were made at 34 μM protein concentration using the same kind of cuvette and spectrometer. A sample of the native protein at pH 7.5 was transformed into an intermediate by heating (92 h at 50 °C) and into amyloid fibrils upon TFE addition to 9% (*v*/*v*) [27] and continuous heating at 50 °C for 45 h. Spectra of the same sample before heating, after heating, and after addition of TFE were measured. Excitation was either 280 nm or 330 nm and the corresponding regions of emission were recorded.

### 4.4. Atomic Force Microscopy (AFM)

The sample was placed on a carrier consisting of a ferromagnetic disk 1 cm in diameter. A double-sided adhesive tape was used to stick a thin layer of freshly cleaved mica, to which 20 µL of the sample solution was applied. After 10 min, during which the fibrils settled, the solution was washed off with 1 mL of microfiltered water and dried with nitrogen flow. Both images in Figure 1 were acquired using Nanoscope IIIa (Digital Instruments), Scanner E in tapping mode (Olympus OMCL-AC160TS, k ~ 30 N/m, f ~ 320 kHz) in ambient air.

### 4.5. Polarized Optical Microscopy

We spread 20 µL of a solution containing 1.5 mM stefin B amyloid fibrils on a glass slide. The sample was analyzed using a Nikon Eclipse E600 POL polarizing optical microscope and a Canon EOS550D camera. In addition, the full-wave plate retarder (λ = 530 nm) was used to determine the local orientation of optical axis.

## Figures and Tables

**Figure 1 ijms-24-03737-f001:**
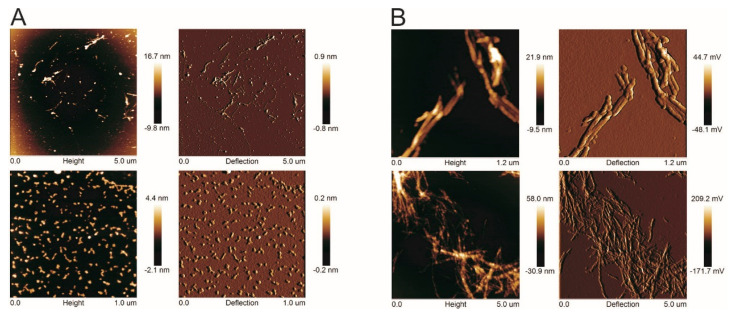
AFM images of stefin B fibrils. AFM images of stefin B fibrils at pH 4.8, room temperature, and 9% TFE (**A**); and pH 7.5, 50 °C, 9% TFE (**B**) are shown at two different magnifications for each sample. For each AFM measurement, height and deflection are shown. Adapted from [25].

**Figure 2 ijms-24-03737-f002:**
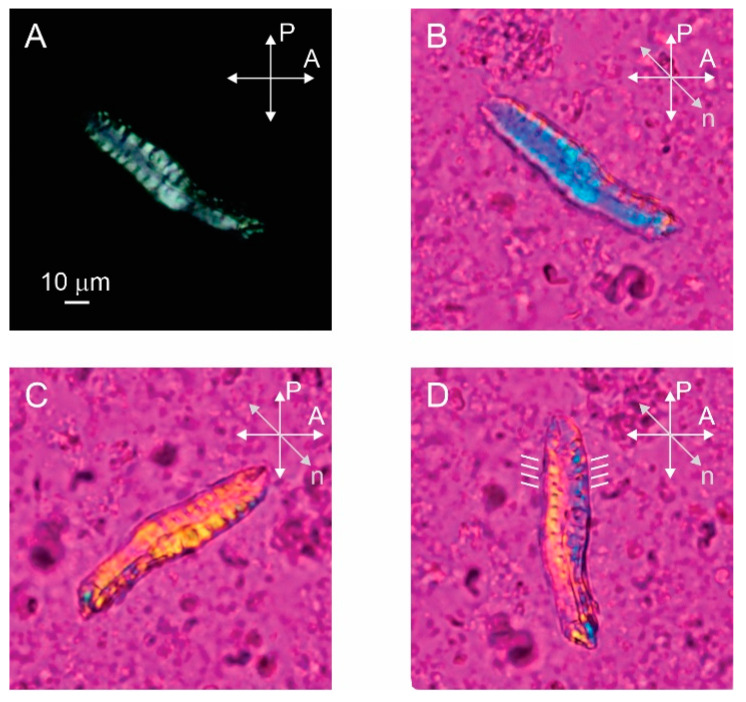
Bundle of stefin B amyloid fibrils prepared at pH 7.5 observed by polarizing optical microscopy. (**A**) Bundle shows birefringence when observed between crossed polarizers. (**B**–**D**) By inserting the red wave plate retarder and rotating the sample we can determine the direction of optical axis and alignment of fibrils, which is indicated by white lines next to the bundle (**D**). P and A show the orientations of the polarizer and the analyser and *n* is the direction of the optical axis of the retarder.

**Figure 3 ijms-24-03737-f003:**
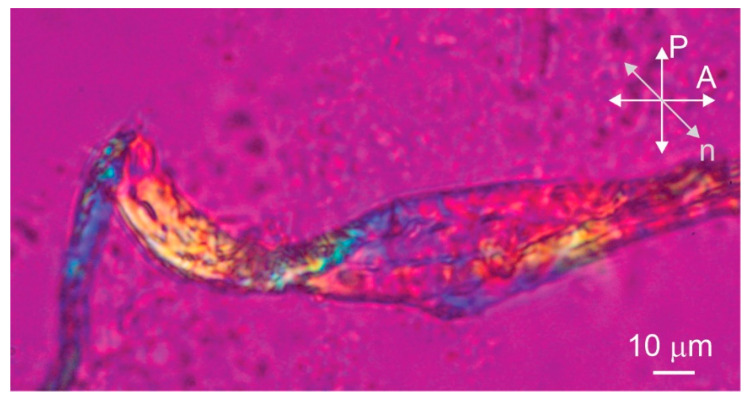
Another entangled bundle of stefin B amyloid fibrils prepared at pH 7.5 observed by polarizing optical microscopy. P and A show the orientations of the polarizer and the analyser, respectively, and *n* is the direction of the optical axis of the retarder.

**Figure 4 ijms-24-03737-f004:**
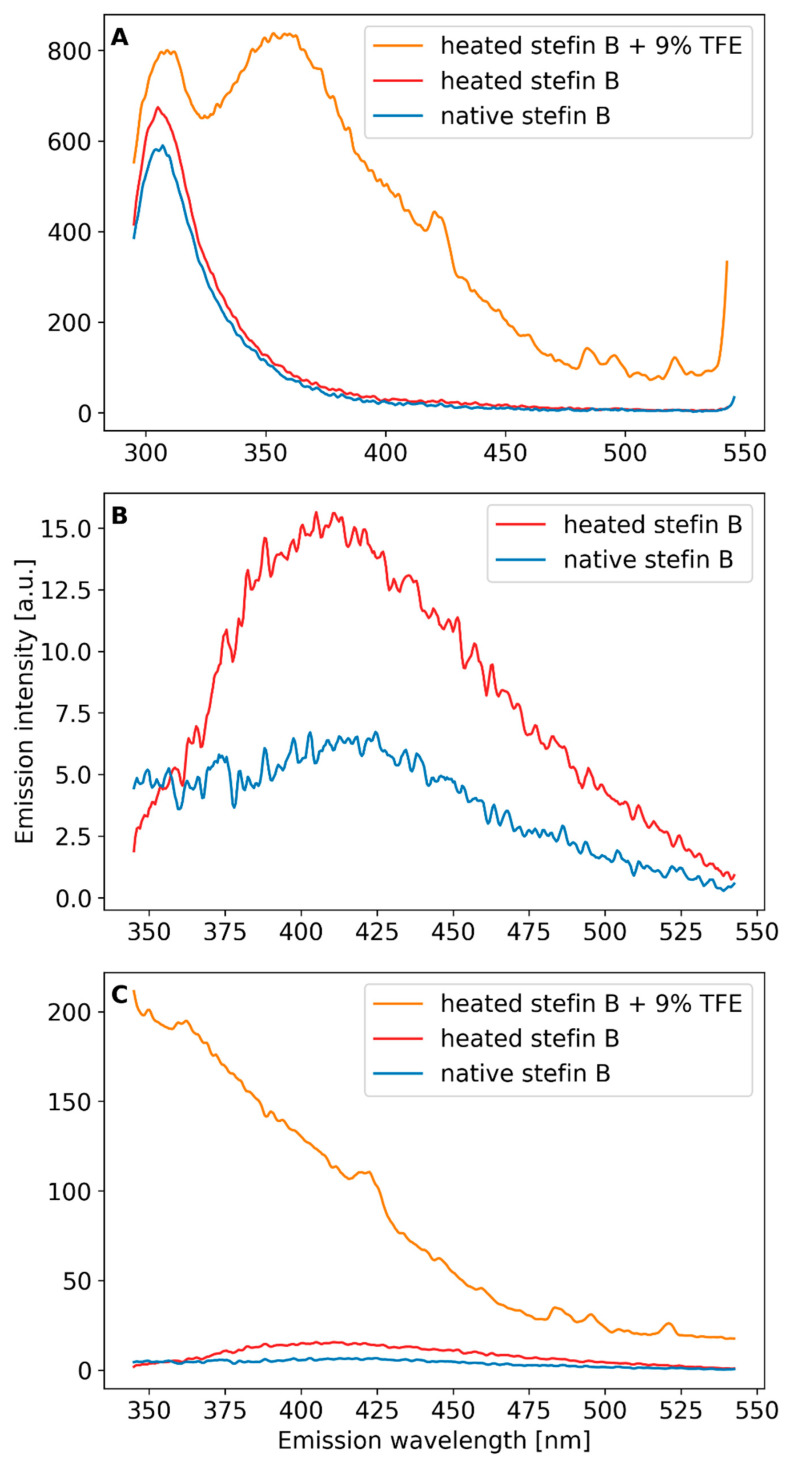
Fluorescence emission spectra (**A**) excitation at 278 nm, PBS buffer pH 7.4, protein conc. 34 µM; sample was heated and TFE added as stated in the curve labels; (**B**,**C**) excitation at 330 nm, fluorescence emission recorded from 345 to 545 nm, pH 7.4, PBS buffer, protein conc. 34 µM; sample of stefin B was first heated (92 h to 50 °C) and 9% (*v*/*v*) TFE was added and the sample heated for another 45 h at 50 °C as indicated in the legend. Heating was always accompanied by shaking at 300 rpm.

## Data Availability

Not applicable.

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
