# Peer review of "Amyloid Fibrils of Stefin B Show Anisotropic Properties"

_ijms, 2023, doi:10.3390/ijms24043737_

Round 1
Reviewer 1 Report
The manuscript is written as a short report that presents new data showing the anisotropic properties of amyloid fibrils formed by human stefin B, which may be used for label-free detection of amyloid fibrils formed in vitro. The data presented is potentially very interesting, but supported only by preliminary data that should be strengthened and complemented with additional experiments (e.g. including amyloid fibrils from different proteins) before publication.
The authors show that stefin B is able to form amyloid fibrils under different conditions and focus on the fibrils formed at pH7.5 at 50°C and in the presence of TFE. Under these conditions, stefin B forms fibrils monitored by ThT fluorescence, which further assemble into fibrillar bundles. Figures 1 and 2 represent the results of experiments previously published bythe authors in 2013 (https://www.mdpi.com/1422-0067/14/9/18362). Most likely the experiments were repeated – why are the data for the new experiments not presented instead?
The figure legends are also unclear. For example, in the legend from Figure 1, it is written – “At certain time points, aliquots were taken and fluorescence was measured. At the plateau of the reactions, aliquots were taken for TEM and AFM”. Which time points were selected? How many replicates were taken for each timepoint? What was the timepoint of the plateau for each condition? Additionally, although in the materials and methods section the experimental conditions for TEM are presented no TEM images are presented in the manuscript.
The new data are presented in Figures 3, 4 and 5. The results show that stefin B fibrils grown at pH 7.5 are able to polarize light in the absence of externally added dyes. The authors then compare the emission spectra of stefin B monomers and fibrils upon excitation at different wavelengths and show a clear emission at 425-430 nm upon excitation within the 300-380 nm range. The results presented, however, do not show convincing differences between monomeric and fibrillar stefin B species. Although the authors state that monomeric stefin B preparations have some oligomers at pH 7.5, further experiments are needed to unambiguously prove that this specific excitation-emission signature is due to the presence of amyloid fibrils in solution.
Finally, the manuscript would benefit from an extensive and careful review of the presentation of the results and discussion. Sentences in the results such as “More on the likely nature of these excitations was explained in the Introduction” should be avoided and a thorough discussion of the results should be presented.
Author Response
Dear Reviewers, see please the attached file. We tried our best to revise the manuscript as much as possible.

Reviewer 2 Report
The manuscript entitled “Amyloid fibrils of stefin B show anisotropic properties” by Žganec M et al. describes the characterization of amyloid fibrils of stefin B. The presented research is a continuation of the previously published work (2013). The manuscript is written clearly, and the experimental procedure is well explained. However, the literature used is largely outdated, and the discussion section is minimal, if not absent. This, unfortunately, makes it very difficult to put the work in perspective and estimate the importance and relevance of the presented research.
Author Response
Dear Reviewer 2, we have gathered all answers in the attached file.

Reviewer 3 Report
In the present study, the authors found that bundles of amyloid fibrils formed by human stefin B exhibit birefringence. The idea that the detection of amyloid fibrils could be possible without any labeling is interesting. However, the interpretations of the results are not convincing at all.
Major:
Optical images in Figure 3 are too huge compared with those in Figure 2. Thus, the amyloid fibrils and bundles seem to be completely different. It is likely that just the bundles not the amyloid fibrils are birefringent. What is the driving force for the formation of the bundles?
In Figure 5, additional peaks around 425-430 nm, if any, look like just a noise because of their low quality of the spectra.
Minor:
3-fluoroethanol should be trifluoroethanol.
At line 101, there is no Figure 2 F,G.
Author Response
Dear Reviewer 3, please see the attached file.

Round 2
Reviewer 1 Report
The authors adequately addressed the comments and suggestions presented in the first review report.
Reviewer 2 Report
Agree
Reviewer 3 Report
The authors cannot discriminate bundles and amyloid fibrils yet.
I do not think that there was a sufficient improvement of the manuscript.